# Flavescence Dorée Phytoplasma Has Multiple *ftsH* Genes that Are Differentially Expressed in Plants and Insects

**DOI:** 10.3390/ijms21010150

**Published:** 2019-12-24

**Authors:** Camille Jollard, Xavier Foissac, Delphine Desqué, Frédérique Razan, Christophe Garcion, Laure Beven, Sandrine Eveillard

**Affiliations:** UMR 1332, INRAE, Université de Bordeaux, F-33140 Villenave d’Ornon, France; camille.jollard@wanadoo.fr (C.J.); xavier.foissac@inra.fr (X.F.); delphine.desque@inra.fr (D.D.); frederique.razan@inra.fr (F.R.); christophe.garcion@inra.fr (C.G.); laure.beven@inra.fr (L.B.)

**Keywords:** phytoplasmas, membrane proteases, protein metabolism, virulence

## Abstract

Flavescence dorée (FD) is a severe epidemic disease of grapevines caused by FD phytoplasma (FDP) transmitted by the leafhopper vector *Scaphoideus titanus*. The recent sequencing of the 647-kbp FDP genome highlighted an unusual number of genes encoding ATP-dependent zinc proteases FtsH, which have been linked to variations in the virulence of “*Candidatus* Phytoplasma mali” strains. The aims of the present study were to predict the FtsH repertoire of FDP, to predict the functional domains and topologies of the encoded proteins in the phytoplasma membrane and to measure the expression profiles in different hosts. Eight complete *ftsH* genes have been identified in the FDP genome. In addition to *ftsH*6, which appeared to be the original bacterial ortholog, the other seven gene copies were clustered on a common distinct phylogenetic branch, suggesting intra-genome duplication of *ftsH*. The expression of these proteins, quantified in plants and insect vectors in natural and experimental pathosystems, appeared to be modulated in a host-dependent manner. Two of the eight FtsH C-tails were predicted by Phobius software to be extracellular and, therefore, in direct contact with the host cellular content. As phytoplasmas cannot synthesize amino acids, our data raised questions regarding the involvement of FtsH in the adaptation to hosts via potentially enhanced recycling of phytoplasma cellular proteins and host protein degradation.

## 1. Introduction

In Europe, Flavescence dorée (FD) threatens viticulture [1,2]. This epidemic disease is caused by a phytoplasma (FDP) of taxonomic subgroups 16SrV-C and 16SrV-D transmitted by *Scaphoideus titanus*, a leafhopper vector of North American origin [3,4]. Symptoms associated to FDP are leaf discoloration (yellowing or reddening) and downward rolling, incomplete lignification of canes and shriveling of grapes and tendrils [5]. FDP infection induces yield reduction and greatly weakens vineyard development [6]. Since no curative treatments can be implemented in vineyards, collective prophylactic control strategies rely on mandatory removal of infected plants, certification of the material for planting, and insecticide treatments against *S. titanus* [7,8]. In the long-term, due to the non-intentional environmental side effects of insecticide use, new innovative methods for the reduction of both the spread and impact of FD are needed. Phytoplasmas are wall-less bacteria of the class *Mollicutes* that exclusively inhabit the phloem tissue of their plant hosts and colonize the insects acting as vectors [9,10,11]. Elucidation of phytoplasma-plant host interactions has long been a challenge, as phytoplasmas have escaped reliable axenic cultivation. Sequencing of phytoplasma genomes has led to major advances in the understanding of the pathogenicity of these minimal bacteria by allowing the identification of secreted proteins that affect plant development and metabolism [12,13,14,15]. Phytoplasmas have a very limited number of gene sets for amino acid synthesis but can import oligopeptides, spermidine/putrescine, and methionine from hosts through ABC transporters [16]. The draft genome of the FDP strain FD92 highlighted the extended gene set for protein degradation, especially genes encoding the ATP-dependent zinc-binding membrane proteases FtsH/HflB [17]. FtsH (filament temperature-sensitive protein) is a membrane-bound ATP-dependent metalloprotease complex found in prokaryotes and organelles of eukaryotic cells [18,19]. In Mycoplasmas, Spiroplasmas, and Acholeplasmas, the three genera belonging to the class *Mollicutes* along with Phytoplasmas, the *ftsH* gene is present as a single copy, with Acholeplasmas being the cultivable mollicute genus that is closely related to the “*Candidatus* Phytoplasma” genus [20,21]. In contrast, four HflB gene copies (syn. FtsH) were identified in the “*Candidatus* Phytoplasma mali” (“*Ca*. P. mali”) linear chromosome [22,23]. Genetic characterization of hypovirulent strains of “*Ca*. P. mali” highlighted the putative role of AAA+ proteins in virulence, particularly FtsH. Accessions of “*Ca*. P. mali” were classified, based on phenotypic symptoms on apple trees, as being strongly, moderately or mildly virulent, and analysis of sequences of *ftsH* and other AAA+ ATPase genes clearly clustered the virulent strains separately from the mildly virulent strains [24]. In addition, mildly virulent strains of “*Ca.* P. mali” can suppress virulent strains in *Catharanthus roseus*, *Nicotiana occidentalis* and apple trees [25]. Some of the virulence traits confer an enhanced ability for plant host colonization to aggressive strains [23,26,27]. The present study aims to describe the gene repertoire for FtsH in the FDP genome, to predict the functional domains and topologies of the encoded proteins, and to establish the expression profile of these genes in different plant hosts and vector insects. We also hypothesize that the expression of *ftsH* genes should be host-dependent. Two pathosystems were challenged, the natural pathosystem consisting of grapevine and *S. titanus*, and an experimental pathosystem in which the phytoplasma strain had been maintained since its transmission to broadbean.

## 2. Results

### 2.1. Eight ftsH Gene Sequences Were Identified in the FDP Genome

For sequence identification, the FtsH protein sequences of *Bacillus subtilis* and *Escherichia coli* were subjected to Blast alignment against the predicted proteome of the FDP strain FD92 [17]. Only the sequences containing FtsH-specific sites were selected, i.e., the two transmembrane domains, ATP-binding sites (walker A and B motifs), pore signature, ATP-hydrolysis site (second region of homology, SRH), zinc-coordinating motif, 3rd Zn-binding glutamic acid residue, conserved aspartic acid, and conserved leucine zipper residues (Figure 1) [19]. The simultaneous presence of all these sites makes it possible to differentiate FtsH from AAA+ ATPases. The eight *ftsH* genes identified in the genome of the FDP strain FD92 had sizes ranging from 1587 bp to 2025 bp, encoding proteins ranging in size from 528 to 674 amino acids and containing all conserved motifs. The genes were numbered according to their position in the chromosome, with *ftsH*1 being the closest to the origin of replication. The distances between specific sites were nearly identical for all eight FtsH proteins, except the distance between the transmembrane domains predicted using Phobius, which varied between 82 and 97 amino acids for six of the proteins and were 20 and 166 amino acids for FTSH7 and FTSH8, respectively (Figure 1). FtsH protein sequences exhibited homology levels ranging from 39% to 75%, and FTSH7 and FTSH1 exhibited the highest level of homology (75%). FTSH6 had the most divergent sequence, with homology to the other FDP FtsH proteins in the range of 39–43%.

### 2.2. Phylogenetic Analysis of Bacterial and Phytoplasma FtsH Proteins Reveals Ancient Duplication of ftsH Genes

Maximum likelihood phylogenetic analysis of bacterial FtsH indicated that all phytoplasma FtsH proteins had a common ancestor (Figure 2). The gene copies were, however, split into two separate clades, one corresponding to what could be considered as the canonic cellular FtsH of “*Ca.* P. mali”, “*Ca.* Phytoplasma ziziphi”, “*Ca.* Phytoplasma australiense”, “*Ca.* Phytoplasma asteris” and FDP, and the other grouping all the other complete paralogs of “*Ca.* P. mali”, “*Ca.* P. ziziphi” and FDP. On one side, clustering of the phytoplasma canonic *ftsH* was consistent with the evolutionary distance between species, as classically illustrated by 16S rDNA phylogeny [9], and indicated *fts*H6 as the original ortholog of the unique bacterial *ftsH*. On the other side, three “*Ca.* P. mali” *ftsH* sequences (ATP_00457, ATP_00034 and ATP_00464, GenBank FR863639, FR863637 and FR863645 respectively), five “*Ca.* P. ziziphi” *ftsH* sequences (WP_121464094.1, WP_121464242.1, WP_121463917.1, WP_121464225.1 and WP_121464146.1), and the seven other FDP *ftsH* paralogs constituted a common phylogenetic branch resulting from ancient duplication events. Interestingly, except for *ftsH*1 and *ftsH*2, which possibly emerged from independent divergent evolution, the other FDP *ftsH* paralogs were clustered according to their position in the FDP chromosome.

The *ftsH* genes are dispersed along the chromosome and are 20 to 81 kbp apart, with six of the genes present on the positive strand (black arrows) and two copies on the negative strand (white arrows) (Figure 3). Therefore, none of these genes were organized into common gene clusters. *ftsH*6 was the only copy surrounded by housekeeping genes (Figure 3 and Appendix A). This gene was preceded by *ser*S (seryl-tRNA synthetase) and *rps*T (ribosomal protein S20) and followed by *rlu*D (23S rRNA pseudouridylate synthetase) and *arg*S (arginyl-tRNA synthetase). The same organization was observed in “*Ca*. P. ziziphi” (group 16SrV) [28], the peanut witches’ broom phytoplasma (group 16SrII) [29], “*Ca* P oryzae” (group 16SrXI) [30] and “*Ca.* P. pruni” (group 16SrIII) [31] around the corresponding *ftsH*6 ortholog. In the evolutionarily distant “*Ca.* P. mali” (group 16SrX), “*Ca.* P. solani” (group 16SrXII) and “*Ca*. P. asteris” (group 16SrI), *lys*S (lysidyl-tRNA Synthetase), *asp*S (aspartyl-tRNA synthetase) and *his*S (histidyl-tRNA synthetase) preceded the ortholog of *ftsH*6 that was followed by only *rlu*D (Appendix A).

### 2.3. FtsH Proteins Are Predicted to Be Differentially Oriented in the Membrane

Many bioinformatic tools are available to predict the membrane spanning topologies of proteins. One benefit of the HMM-based Phobius server is the separate models for cleavable signal peptides and transmembrane (TM) segments [34]. Seven *ftsH* genes identified in the FDP genome encode proteins with two TM segments identified using Phobius (Table 1).

This feature is common to the unique FtsH found in bacteria other than phytoplasmas. For FtsH2, a signal peptide and a TM segment were predicted. Cleavage of the signal peptide sequence cannot be predicted in silico, as it has been formerly shown that phytoplasma membrane proteins such as IMP can possess typical signal peptides while remaining anchored at the outer surface due to lack of cleavage [35]. For most of the FtsH proteins, the *C*-tail was predicted to be intracellular, with the exception of FtsH6 and FtsH7. For these two proteins, the protein region bearing the potential active sites were thus predicted to be surface-exposed, that is, in direct contact with the host. This method was applied to 11 other publicly available unique FtsH sequences from *Acholeplasma* species, which are phylogenetically close to Phytoplasma (data not shown). These proteins were all predicted to face the cytoplasm, which is consistent with the described topology of the canonical bacterial FtsH. However, the number of predicted TM segments and topology models were different depending on the predictor used. Specifically, depending on the FtsH, 1 or 2 TM segments were predicted using TMHMM. Thus, a detailed analysis of the membrane topology models was performed.

In the Phobius membrane topology model, the large part of FtsH containing the functional sites has a well conserved length among the 7 FtsH proteins (438–460 amino acids). The predicted soluble part of FtsH6 is longer (538 amino acids) and contains a specific sequence at the C-terminus, presenting similarities with *Acholeplasma* FtsH. In these models, FtsH differed in the length of Loop1 between TM1 and TM2, from 20 amino acid residues for FtsH7 to 166 amino acids for FtsH8. As discussed further, the net charge of the extra membrane loops and the density of positively charged lysine (K) and arginine (R) may have an effect on membrane topology. The net charge of Loop1 varies between −1 and +6; the percentage of positive residues is also significantly different, with 14.4% for Loop1 in FtsH1 and 20% for FtsH7. TM2 of FDP FtsH exhibits a noticeable specificity compared to TM2 of *Acholeplasma* FtsH, with enrichment of polar ionizable amino acids in the core of the TM segment. Most of these amino acids are neutral (C, S, Y, Q, N), but five FtsH proteins also have charged acidic (D) or basic (K, R) residues in their TM2 sequences. FDP FtsH characteristics such as the presence of charged residues in putative TM segments and differences in sequences corresponding to the Phobius predicted Loop1 net charge led to the prediction of a single TM segment in FtsH2, 3, 4, and 7 using TMHMM.

Thus, a reliable membrane topology model for each FtsH could not be obtained due to contradictions among different topology predictors. Nevertheless, both predictors argue in favor of different topologies depending on the FtsH sequence, the largest region of some being surface-exposed and those of others facing the cytoplasm.

### 2.4. Substrate-Binding Proteins Are Encoded by Genes Upstream of Several ftsH Genes

Analysis of the *ftsH* gene environment revealed the occurrence of genes encoding homologous proteins upstream of the complete *ftsH* genes in the same operon, with the exception of *ftsH*2 and *ftsH*6. The shortest genes were considered to encode truncated proteins. The four longest proteins (from 419 aa to 720 aa) were encoded by genes preceding *ftsH*1, *ftsH*3, *ftsH*4 and *ftsH*5. For these four proteins (in grey in Figure 3B), internal sequences corresponding to functional domains of the substrate-binding unit of bacterial ABC transporters for oligopeptides were identified using the NCBI Blast automatic domain detection tool. Proteins encoded by genes upstream of *ftsH*1, 3, and 5 also contained a predicted signal peptide for membrane translocation, while the protein encoded by the gene upstream of *ftsH*4 appeared to be truncated in the N-terminal region (Figure 3B). Such a genetic association between a substrate-binding related subunit gene and an *ftsH* gene was also identified in other phytoplasmas, such as “*Ca*. P. mali” (loci ATP_00456 and ATP_00457 of Genbank accession) and “*Ca*. P. pruni” (loci CPX_001389 and CPX_001388).

### 2.5. Phytoplasma ftsH Genes Are Differentially Expressed Based on the Host

To evaluate the potential influence of host diversity, *ftsH* gene expression was analyzed in two different pathosystems: a natural system consisting of FDP-infected grapevine and the natural vector *S. titanus*, and an experimental system consisting of FDP-infected broad bean and the Cicadellidae family member *Euscelidius variegatus*. Because most functional studies are done on this experimental system, it appeared of interest to determine the gene expression both in the natural and in the experimental pathosystems. Indeed, comparing the two pathosystems allowed us to determine if there were any differences in the expression of *ftsH* genes in the natural hosts as compared to the experimental ones. The presence of FDP in the different samples was verified by a qPCR test [36], and the determined phytoplasma titers varied between 1.07 × 10^+05^ and 2.17 × 10^+05^ phytoplasma/mgFW in grapevine (mean 1.30 × 10^+05^), 2.83x10^+06^ and 5.27 × 10^+06^ phytoplasma/insect in *S. titanus* (mean 4.11 × 10^+06^), 1.13 × 10^+06^ and 3.74 × 10^+06^ phytoplasma/mgFW in broad bean (mean 2.56 × 10^+06^), and 2.65 × 10^+04^ and 1.4 × 10^+05^ phytoplasma/insect in *E. variegatus* (mean 5.79 × 10^+04^). Healthy samples, either plant or insect, did not show any amplification.

The expression of all *ftsH* genes was tested by RT-PCR for both pathosystems with specific primers (Table 2). No amplification was observed in the different controls, healthy hosts and non-reverse transcript RNA samples (not shown). The results showed that all *ftsH* genes but one were expressed in FDP-infected grapevine, broad bean, *S. titanus* and *E. variegatus* (Figure 4); *ftsH*8, even if expressed in the plant hosts, was negligibly expressed in the insect hosts.

The expression profiles were similar between the two insect species but a little different between the two plants. In grapevine, the most highly expressed *ftsH* genes were *ftsH*3 and 5, and the weakest-expressed ones were *ftsH*1, 2, and 8. In broad bean, *ftsH*3 and 5, but also 6, were the most highly expressed, while *ftsH*4 and 8 were the most weakly expressed as compared to the others, which was not observed in grapevine. When comparing *ftsH* gene expression levels between grapevine and broad bean, the results showed that 3 *ftsH* genes out of eight have comparable levels of expression (*ftsH*1, 2, and 6), while the other genes have higher levels of expression in grapevine than in broad bean.

The results in insects were different from those observed in plants. In *S. titanus,* the most highly expressed *ftsH* genes were *ftsH*2, 5, 6, and 7. Under these conditions, *ftsH*4 was expressed at low levels, while *ftsH*8 was negligibly expressed. The expression profile in *E. variegatus* was similar to that observed in *S. titanus*. *ftsH* gene expression was compared between *S. titanus* and *E. variegatus*, and 5 genes exhibited similar levels of expression (*ftsH*1, 2, 3, 5, and 6), while only two exhibited lower levels of expression in *S. titanus* than in *E. variegatus;* we considered *ftsH*8 to not be expressed in insects.

These results showed that there were not many differences between the natural and experimental pathosystems in terms of the expression of *ftsH* genes, while there exists a distinct difference in expression between plants and insects.

### 2.6. Three ftsH Genes Are More Expressed in Grapevine than in S. titanus

Relative gene expression (RGE) was calculated to compare *ftsH* expression in two different hosts (Figure 5). The results showed that *ftsH*3, 4 and 8 were more highly expressed when the FDP was hosted in grapevine than when the FDP was in *S. titanus*. We have previously noted that *ftsH*8 was negligibly expressed in *S. titanus*. *ftsH*3, one of the most highly expressed *ftsH* genes in grapevine, and *ftsH*4, which is moderately expressed in grapevine, were 10 times more highly expressed when the FDP was in grapevine than when the FDP was in *S. titanus*. In contrast, *ftsH*1, 2 (weakly expressed), and 6 (moderately expressed) were more highly expressed when the FDP was in *S. titanus* than when the FDP was in grapevine. The difference observed for *ftsH*7 was not statistically significant. *ftsH*5, which is highly expressed in all hosts, was equally expressed in grapevine and in *S. titanus*. These results indicated that except for *ftsH*5, a shift in expression between *S. titanus* and grapevine was observed for the other *ftsH* genes.

### 2.7. ftsH8 is More Expressed in Broad Bean than in E. variegatus

The RGE determined for the *ftsH* genes in the experimental pathosystem differed from that observed in the natural pathosystem. Only one *ftsH* gene (*fstH*8) was more highly expressed in broad bean than in *E. variegatus*. Five *ftsH* genes (1, 2, 5, 6, and 7) were more highly expressed in *E. variegatus* compared to broad bean*,* and 2 *ftsH* genes (3 and 4) were almost equally expressed in these two hosts (Figure 5). The results for *ftsH*3, 4, 5, and 7 were distinctly different from those obtained in the natural pathosystem. Indeed, *ftsH*3 and 4 were overexpressed in grapevine compared to *S. titanus*, but not in broad bean, and *ftsH*5 and 6 were nearly equally expressed in the natural pathosystem, which is not the case for the experimental pathosystem.

## 3. Discussion

Eubacterial FtsH is an ATP-fueled metalloprotease that presents a membrane-bound hexameric structure that is able to unfold and degrade protein substrates [37]. While a single gene encodes FtsH in most bacteria, eight copies containing all the characteristic features of *ftsH* were identified in the FDP genome. Such a high number is a noticeable feature because phytoplasmas have small genomes that mainly evolve via gene loss. It is then expected that these highly duplicated genes may have an essential function either for survival or for pathogenicity. A high number of paralogs are also found in other phytoplasmas: Four in “*Ca*. P. mali” and six in “*Ca*. P. ziziphi” [16]. It was proposed that in the absence of the CRISPR system, the multiplicity of *ftsH* paralogs could be linked to a high pressure of phage attacks and reflect the necessity of defense, as in *E. coli,* where FtsH (HflB) plays a potential role in lambda phage lysogeny [16,38]. Other phytoplasmas possess several copies of membrane AAA+ ATPases in their genomes. These copies can be found in potential mobile units (PMUs), such as in three PMUs out of four in the “*Ca*. P. asteris” AY-WB strain and in the 4 PMUs of jujube witches’ broom phytoplasma [13,28]. These PMUs are organized as clusters with many proteins potentially involved in the interaction of ‘*Ca*. P. asteris’ AY-WB with plant and insect hosts. The regulation of the expression of PMU genes is proposed to be one of the strategies used by phytoplasmas to adapt to different environments [13]. Thus the elevated numbers of AAA+ /FtsH in phytoplasma genomes can be in some cases explained by the duplications of PMUs. In the FDP genome, however, *ftsH* genes do not seem to be in PMUs and, unlike membrane AAA+ ATPases, the encoded proteins do not lack the zinc-binding and protease domains [17].

The possible functions of each FDP FtsH protein depends on the topology of the protein in the phytoplasma membrane. A clear model for FDP FtsH membrane topology could not be obtained. This issue could be attributed to the lack of phytoplasma protein sequences in the set of proteins used to construct the hidden Markov models. Indeed, several characteristics of phytoplasma membrane proteins have been described in the literature, such as the absence of the cleavage of predicted signal peptides [35], suggesting that the secretion system likely bears some specific features in phytoplasmas compared to other bacteria, including other mollicutes. However, two conclusions could be extracted from the in silico analyses reported in our work: (1) analysis of Phobius topology models indicate that all FDP FtsH proteins have all the putative functional sites of comparable sizes in the C-terminal region; (2) prediction by Phobius and TMHMM argue in favor of different membrane topologies depending on the FtsH. The co-occurrence of oppositely oriented transmembrane FtsH proteins has also been hypothesized for “*Ca*. P. mali” FtsH [24]. This feature in FDP as well as in other phytoplasmas requires plasticity in the mechanisms which regulate the bacterial membrane protein insertion mechanisms. Such plasticity has been described in other biological models such as *E. coli* where dual-topology membrane proteins such as the transporter EmrE in *E. coli* can adopt both orientations. Addition of some positive residues (K and R) at selected positions in the sequence, even in the *C*-terminal part, can invert the protein insertion [39]. This observation is consistent with the positive-inside rule, which states that positive residues prefer the cytoplasmic side [40]. The positive charge distribution differs among FDP FtsH, possibly allowing different orientations of the proteins. Moreover, a divergent evolution of topology for polytopic strongly related proteins by reshuffling of positively charged residues has also been reported [41]. Finally, two FtsH homologues, namely, *i*-AAA (two membrane spanning segments) and *m*-AAA (three transmembrane segments), present in the inner membranes of mitochondria have inverted topologies [42]. The present study suggests that in FDP, evolution would have imparted a different membrane topology for some FtsH paralogs compared to other bacterial FtsH proteins.

FtsH with cytoplasmic catalytic sites may play a role similar to those identified for the eubacterial universal FtsH known to catalyze the degradation of denatured, misassembled or damaged proteins and enable cellular regulation at the level of protein stability [37]. The multiple FDP FtsH proteins facing the phytoplasma cytoplasm may then be associated with protein quality and a short turn-over, which is associated with decreased reliability of the cellular machinery under stress conditions. The cellular function of FtsH facing the extracellular medium is likely to be different than that of intracellular ones. Approximately half of the “*Ca*. P. mali” FtsH proteins were predicted to have extracellular active sites, suggesting an involvement of these proteins in pathogen-host interactions [23].

Genes encoding proteins that are homologous with the substrate-binding protein (SBP) of bacterial ABC transporters for oligopeptides were identified in 4 *ftsH* operons. Three of these proteins are likely to be translocated through the membrane and expressed as surface-exposed polypeptides in their mature form. One may thus hypothesize that the extracellular FtsH proteases and the transporter units could act together to allow the exchange of peptides between the host environment and the FDP cell. Following this assumption, the change in membrane orientation of FtsH would be associated with a gain in capacity for peptide/amino acid import, an essential function for phytoplasmas that are devoid of enzymes required for amino acid biosynthesis. However, bacterial ABC transporters are composed of subunits whose association allows substrate binding and translocation, which is coupled to ATP hydrolysis [43]. Depending on the transporter, some subunits can perform one or two required activities, such as substrate binding and permease activity. SBP proteins identified in this work could bind peptides, and all the FtsH proteins bear all motifs required for ATP hydrolysis. These findings raise questions regarding the identification of the permease unit. On the one hand, no gene in the surroundings of *ftsH* could be identified as encoding a permease unit. Moreover, no oligopeptide permease could be identified in the FDP genome, in contrast to other members of the taxonomic group 16SrV, such as “*Ca.* P. ziziphi” that possesses a complete oligopeptide ABC transporter [17,28]. On the other hand, TM2 in FDP FtsH showed specific features, including the presence of many polar residues, including charged residues for several FtsH proteins. The presence of such residues in TM segments could allow a change in helix packing in FDP FtsH compared to canonical bacterial FtsH.

All *ftsH* genes seem to be expressed in plants and insects, except for the *ftsH*8 gene, which is very poorly expressed in insects, suggesting that this gene might play a role in only plant hosts. These results are consistent with a RNAseq study of a different FD-P strain in field-grown grapevines that showed the expression of five different *ftsH* genes [44]. The expression profiles are slightly different in grapevine and broad bean, *ftsH*3 is the most highly expressed for both hosts, but differences are observed mainly for *ftsH*4, 6, and 8. *S. titanus* and *E. variegatus* show similar profiles, with *ftsH*5, 6, and 7 being the most highly expressed. This finding indicates that there is no specificity of expression for the insects, either natural *S. titanus* or experimental *E. variegatus*, while there is specificity of expression for the plants, grapevine and broad bean as expression profile is not the same. One could consider that FDP has to adapt its FtsH machinery (i) to variations of its own proteinic composition associated with host change and (ii) to variations in proteinic composition of the different hosts. One way to achieve such adaptation is to express different FtsH with different specificities. Such a differential regulation was also observed in the “*Ca*. P. asteris” OY-M strain, where two membrane AAA+ ATPases (PAM064 and PAM695) were shown to be more highly expressed in the insect vector *Macrosteles striifrons* than in the host plant *Chrysanthemum coronarium*, while two others (PAM117 and PAM480) were equally expressed [45]. Since AAA+ ATPAse and HflB/FtsH have been postulated to be associated with virulence, it is possible that different sets of FtsH are more specifically adapted to one host or the other. Indeed, host-dependent expression could also be related to phytoplasma capability to respond to the stress imposed by the host-cell environment. In *Mycobacterium tuberculosis, ftsH* expression was upregulated upon exposure to agents producing reactive oxygen and nitrogen intermediates, and overexpression of *ftsH* in this bacterium increases resistance to reactive oxygen species and intermediates, while *ftsH* transcript levels are downregulated during the stationary phase and starvation [46]. A similar role was also suggested for FtsH when *B. subtilis* was subjected to environmental stress [46,47]. FtsH/HflB was also found to be involved in temperature and salt tolerance in *B. subtilis*, *Lactococcus lactis*, and *Lactobacillus plantarum* through degradation of denatured proteins [46,48,49,50]. It is possible that either plant or insect response to phytoplasma infection leads to an increase of oxidative stress towards phytoplasma which could, in turn, induce modifications of *ftsH* expression to enhance protection. This could be associated with phytoplasma-superoxide dismutase expression as shown for “*Candidatus* Phytoplasma asteris” OY-W strain and “*Candidatus* Phytoplasma mali” which possess the *sod2* gene encoding a protein that can inactivate reactive oxygen species [51]. Phytoplasma infection in plants is associated with the propagation of a Ca2+ influx, metabolic shift and structural reorganization in the sieve element [52,53,54,55,56]. It is likely that such changes in the host plant most likely induce a complex physiological response of phytoplasmas that, in addition to the classic shift in gene expression, would require additional supply and recycling of amino acids that could imply FtsH.

Considering all these data, two main models for the function of extracellular-exposed FtsH could be proposed: in the first model, FtsH facing the extracellular medium could participate in peptide import by allowing ATP-driven substrate translocation toward the cytoplasm. In the second model, FtsH would act as a protease that degrades host proteins, either for protection of FDP against harmful components, such as plant PRs or insect antimicrobial peptides, or for use of peptides as nutrients. In the latter case, these peptides would be transported by an unidentified transmembrane protein (Figure 6). Functional studies aimed at characterizing the proteolytic, ATPase, and transport activities of FtsH. Identifying the degraded proteins are now required to better understand the roles of FtsH proteins in FDP pathogenicity and fitness.

## 4. Materials and Methods

### 4.1. Phytoplasma Isolates, Plants, and Insects

FD is a quarantine disease subjected to mandatory removing of infected grapevine in vineyard. Moreover, the natural insect vector, *S.titanus* cannot be grown in colony. In order to infect grapevine with FDP, we used an experimental pathosystem as a source for FDP. The FDP strain FD92 was transmitted to broad bean (*Vicia faba*) by *S. titanus* leafhoppers collected in 1992 from an FDP-infected grapevine in southwestern France [57]. This strain has been maintained by serial transmission from broad bean to broad bean by *Euscelidius variegatus* as an experimental leafhopper vector, as FDP is not cultivated [58]. This strain is transmissible by the natural vector *S. titanus* and produce symptoms similar as those observed in vineyards. *E. variegatus* were reared on oat and broad bean while *S. titanus* natural vectors was reared on grapevine (Cabernet Sauvignon and Riparia Gloire de Montpellier). After hatching, *S. titanus* were infected by feeding on infected broad bean for six days (125 insects per broad bean showing symptoms) followed by a latency period of 4 weeks on grapevine (Cabernet Sauvignon) under confined greenhouse conditions [36]. After this latency period, seven *S. titanus* insects were encaged on each two months old healthy grapevine for one week to obtain inoculated grapevines. Leaves of three FDP-infected grapevines were collected seven weeks post inoculation as well as leaves of three healthy grapevines of the same age. Leaves were also collected four weeks post-inoculation on three healthy and three infected broad bean. Three *S. titanus* and 3 *E. variegatus* were collected four weeks after feeding on an infected grapevine or broad bean respectively, as well as three *S. titanus* and three *E. variegatus* of the same age collected on healthy grapevine or broad bean, respectively.

### 4.2. Nucleic Acid Extraction

Petioles of 5–7 leaves of each plant were ground to a fine powder in liquid nitrogen. Approximately 50 mg of this powder was used for DNA extraction using the protocol described in [59] with a final resuspension of the pellet in 30 µL of TE1X (10 mM Tris-HCl, 1 mM EDTA). This DNA was used to determine phytoplasma titers. Total grapevine and broad bean RNA was extracted from 100 mg of fine powder using the Spectrum Plant Total RNA Kit (Sigma-Aldrich, St. Quentin Fallavier, France) with some modifications, as described in [60], and the RNeasy Plant Mini Kit (Qiagen, Hilden, Germany), respectively. Ten insects were ground directly in TRIzol reagent, and DNA and total RNA were isolated separately following the protocol described by the manufacturer. All RNA samples were treated with RQ1 DNase (Promega, Madison, WI, USA) following the manufacturer’s recommendations. RNA extracts were analyzed in a NanoVue^TM^ Plus (GE Healthcare, Buc, France) to evaluate concentration and purity.

### 4.3. Phytoplasma Quantification

The presence of phytoplasma in the different host samples was verified by qPCR using the protocol described by Eveillard [36] and expressed in phytoplasma/mg fresh weight (mg FW) for plants or in phytoplasma/insect.

### 4.4. In Silico Sequence Analysis of FtsH

Genes encoding putative FtsH protein were identified in the FDP genome using Blast with FtsH protein sequences of *Bacillus subtilis* and *Escherichia coli* as queries against the predicted FDP proteome [17]. FDP FtsH sequences were aligned with eubacterial FtsH sequences using ClustalW [61] to identify conserved motifs and domains. The AAA+ ATPases were removed from the analysis. The Phobius predictor [62] was used for prediction of transmembrane segments. Nevertheless, the results obtained with these predictors for topology were compared to those obtained with other prediction tools, including TMHMM [63] and SignalP 4.0 [64].

### 4.5. cDNA Synthesis and Quantitative Real-Time Reverse Transcription PCR (RT-PCR)

For each sample of grapevine, broad bean, *E. variegatus* and *S. titanus*, 1 µg of RNA was used for cDNA synthesis using Superscript Reverse Transcriptase III and random primers following the manufacturer’s recommendations (Invitrogen, Vilnius, Lithuania). The final pellet was resuspended in 20 µL of RNase-free water. Real-time RT-PCR was performed using 1 µL of cDNA template in 1× Light Cycler 480 SYBR Green Master Mix (Roche, Mannheim, Germany) with 0.1–1 µM each primer in a final volume of 20 µL (Table 2). The reaction conditions were as follows: 15 min at 95 °C, followed by 36 or 40 cycles of 20 s at 95 °C; 30 s at 56, 60, 62, or 64 °C; and 30 s at 68 °C (Table 2). The specific primers designed to specifically amplify each *ftsH* gene are presented in Table 2 with the corresponding amplification conditions. The efficiency of RT-PCR varies between 90% and 100%, and all primers used show a single melting peak temperature between 74.1 °C (*ftsH*3) and 78.4 °C (FD-*gyrA*).

The three biological replicates (plants or insects) were run in duplicate on each of the three experimental replicates (plates) comprising negative and positive controls. Amplification of the DNA dilution series was performed to calculate the amplification efficiencies for each gene assay.

### 4.6. Data Analysis

Gene expression was normalized with two reference genes (*gyrA* and *dnaB*). Values are expressed as log MNE (mean normalized expression) [65] or as relative gene expression (RGE) [66]. MNE allows the comparison between samples as it normalizes the expression with two phytoplasma reference genes. Indeed, expression of different *ftsH* can be compared in one host. RGE allows the comparison of different *ftsH* expression between two hosts.
MNE = (E_ref_)^Ct (ref, mean)^ / (E_target_)^Ct (target, mean)^(1)
(2)$$\mathrm{RGE}\;=\frac{E_{\mathrm{target}}^{\mathrm{Ct}\;\left(S.\mathrm{titanus}\;\mathrm{or}\;E.\mathrm{variegatus}\right)\;-\;\mathrm{Ct}\;\left(\mathrm{grapevine}\;\mathrm{or}\;\mathrm{Broad}\;\mathrm{bean}\right)}}{E_{\mathrm{ref}}^{\mathrm{Ct}\;\left(S.\mathrm{titanus}\;\mathrm{or}\;E.\mathrm{variegatus}\right)\;-\;\mathrm{Ct}\;\left(\mathrm{grapevine}\;\mathrm{or}\;\mathrm{Broad}\;\mathrm{bean}\right)}}$$
where E is the rt-PCR efficiency; Ct is the cycle threshold; target is the target gene (*ftsH*); and ref is the reference gene (*gyrA* and *dnaB*)

MNE were submitted to Shapiro test for normality. Results oriented us towards the non-parametric Wilcoxon test to compare MNE, giving the letter of Figure 4. The same statistical analysis was done for RGE (Figure 5).

### 4.7. Molecular Phylogenetic Analysis by the Maximum Likelihood Method

The eight FDP *ftsH* genes were deposited in the European Nucleotide Archive under the accession numbers LT999755 (FTSH1) to LT999762 (FTSH8). The evolutionary history was inferred using the maximum likelihood method based on the JTT matrix-based model [32]. The tree with the highest log likelihood (−17,027.0586) is shown. The percentage of trees in which the associated taxa clustered together upon 250 bootstraps is shown next to the branches. Initial tree(s) for the heuristic search were obtained by applying the neighbor-joining method to a matrix of pairwise distances estimated using a JTT model. The tree is drawn to scale, with branch lengths measured in the number of substitutions per site. The analysis involved 42 amino acid sequences. All positions containing gaps and missing data were eliminated. There were a total of 417 positions in the final dataset. Evolutionary analyses were conducted in MEGA6 [33].

### 4.8. Sequence Data

Sequence data from this article can be found in the European Nucleotide Archive databases under the studies accession number PRJEB26323.

## Figures and Tables

**Figure 1 ijms-21-00150-f001:**
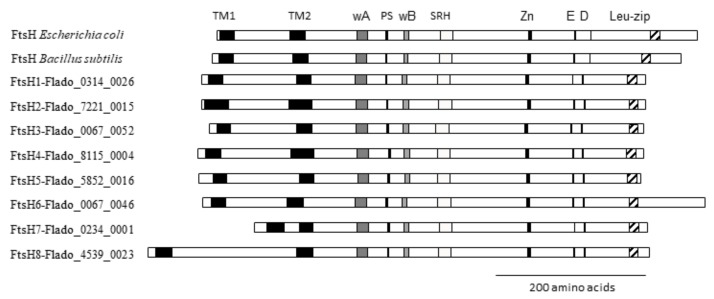
FtsH proteins deduced from the eight *ftsH* genes of the genome of the FDP strain FD92 with the different functional domains characteristic of FtsH: two predicted transmembrane domains (TM1 and TM2), ATP-binding sites walker A and walker B (wA and wB), pore signature (PS), ATP-hydrolysis site (SRH), the HExxH zinc-binding motif (Zn), the third Zn-ligand-binding glutamic acid residue (E), the third coordinating aspartic acid residue important for the proteolytic activity (D) and a leucine-zipper residue for substrate binding (Leu-zip). The *E. coli* and *B. subtilis* FtsH proteins are shown as references.

**Figure 2 ijms-21-00150-f002:**
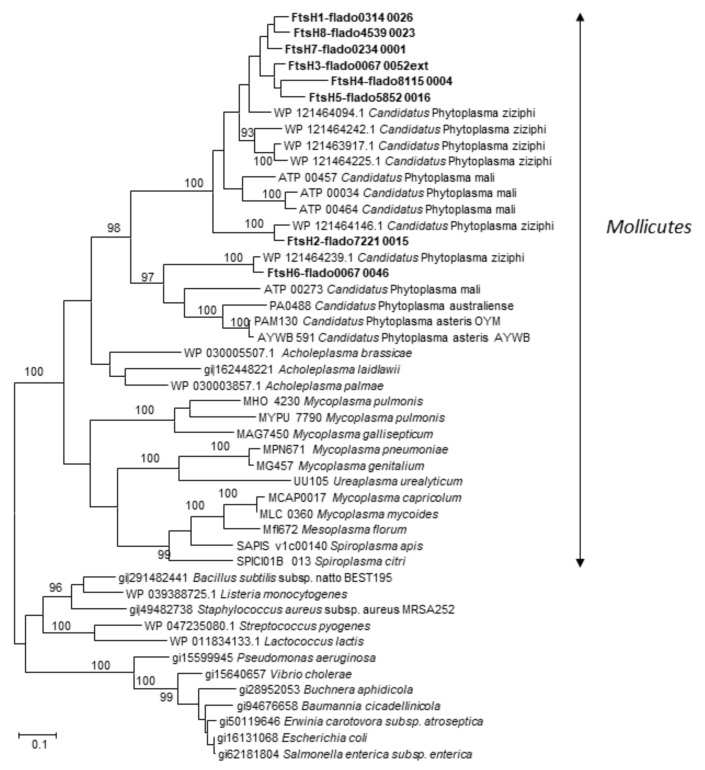
Molecular phylogenetic analysis by the maximum likelihood method based on the proteic sequences [32]. The analysis involved 42 amino acid sequences. Evolutionary analyses were conducted in MEGA6 [33]. The 8 FtsH from FDP are in bold. The tree is drawn to scale, with branch lentghs measured in the number of substitutions per site.

**Figure 3 ijms-21-00150-f003:**
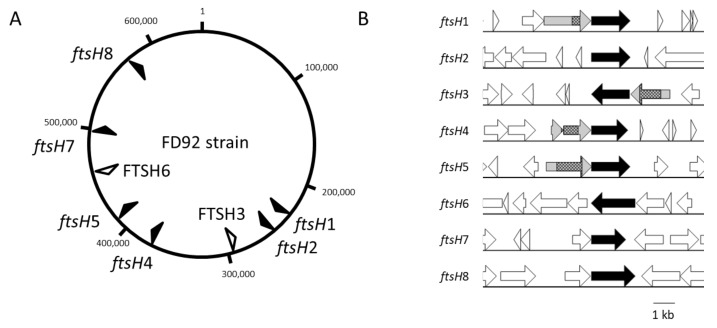
(**A**) Position of *ftsH* genes on the FDp genome (strain FD92). Black arrows indicate CDS on the (+) strand, and white arrows indicate CDS on the (−) strand. (**B**) Genomic context of *ftsH* genes. CDS on the (+) strand are represented with a right arrow, and those on the (−) strand are represented with a left arrow. Black arrows correspond to the FtsH CDS, hatched boxes to domains with similarity to ABC transporters substrate-binding proteins, and gray arrows to CDS associated with a putative transporter function.

**Figure 4 ijms-21-00150-f004:**
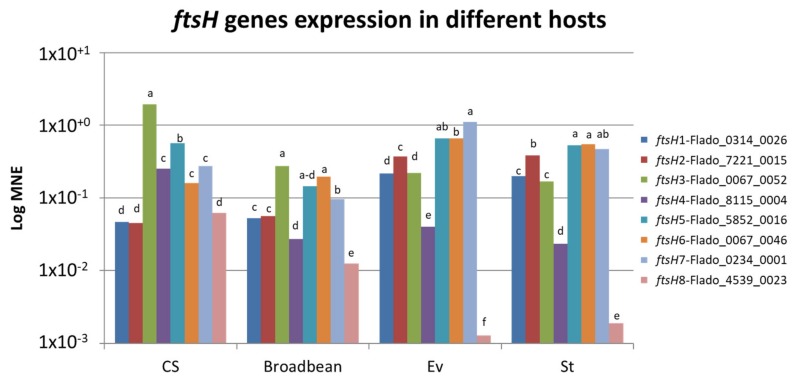
Relative *ftsH* gene expression (log mean normalized expression (MNE)) of FDP strain FD92 in plant hosts (CS: grapevine and broad bean) and in insect vectors *(Ev: E. variegatus* and St: *S. titanus*). Different letters refer to different mean values calculated for each host (Wilcoxon test, *p* < 0.05).

**Figure 5 ijms-21-00150-f005:**
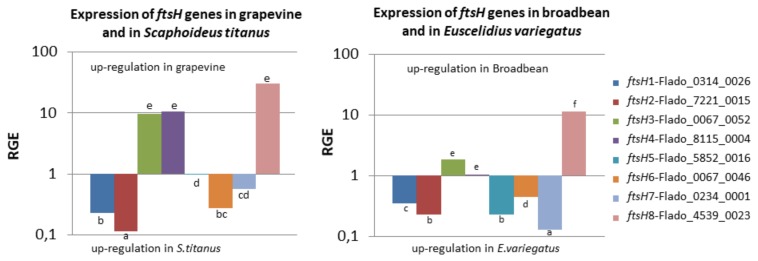
Relative gene expression (RGE) of *ftsH* genes in infected grapevine and leafhopper vector *S. titanus* (left) and in infected broad bean and leafhopper vector *E. variegatus* (right). Different letters refer to different mean values (Wilcoxon test, *p* < 0.05).

**Figure 6 ijms-21-00150-f006:**
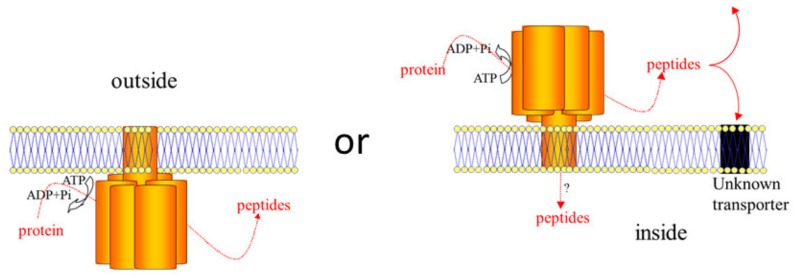
Schematic representation of possible insertion of FtsH in the FD phytoplasma membrane. FtsH are represented as hexamers anchored in the phytoplasma membrane. The C-tail is facing the inside of the phytoplasma (left) or the outside (right). When facing the inside, FtsH can degrade protein into peptides. When facing the outside, FtsH can degrade host’s proteins and peptides can either stay in the outside of the phytoplasma, or can be imported in the phytoplasma via an unknown way.

**Table 1 ijms-21-00150-t001:** Prediction of the C-terminal part of the FtsH protein orientation using Phobius software. *: one signal peptide and one transmembrane domain predicted.

Protein	Number of Amino Acids	Number of Transmembrane Domains Predicted	C-tail Orientation (Phobius), out (%)
FtsH1-Flado_0314_0026	595	2	9
FtsH2-Flado_7221_0015	596	2	8
FtsH3-Flado_0067_0052	584	2	10
FtsH4-Flado_8115_0004	599	2	1
FtsH5-Flado_5852_0016	611	2	3
FtsH6-Flado_0067_0046	674	2	74
FtsH7-Flado_0234_0001	528	2	86
FtsH8-Flado_4539_0023	673	2	25

**Table 2 ijms-21-00150-t002:** qPCR primer sequences for *ftsH* and control genes, primer concentration used, amplicon length and melting temperature, number of cycles and efficiency of qPCR. FD-*gyrA* corresponds to Flado_5136_0003 and FD-*dnaB* to Flado_5892_0007.

Gene Name			Final Concentration (µM)	Length (bp)	Tm (°C)/Number of Cycle	Efficiency
*ftsH*1	Forward primer 5′–3′	TTCTGAATTTGTTGAAATGTACG	1	156	60/36	100%
	Reverse primer 5′–3′	TTTTCTTGAGATCCTCCTGAGA				
*ftsH* 2	Forward primer 5′–3′	TAGCTGGGGAATCAGGAGTT	0.1	553	60/40	90%
	Reverse primer 5′–3′	ACAGCTTCTTCTAATTCTATCAT				
*ftsH* 3	Forward primer 5′–3′	TGGTTCTGAATTTATGGATAGA	1	157	62/36	99%
	Reverse primer 5′–3′	GTACTTCCATCATCATTTGC				
*ftsH* 4	Forward primer 5′–3′	AATCGGCATTAAACTCCCTCG	1	207	60/36	100%
	Reverse primer 5′–3′	TCATAAGCACCACTAGAAACT				
*ftsH* 5	Forward primer 5′–3′	GTGTTGGTGCTTCTCGTGTC	0.1	291	60/36	90%
	Reverse primer 5′–3′	CACGTGCTTTAACATCTGGC				
*ftsH* 6	Forward primer 5′–3′	GTTTTGGACCTCTTTTATCTGC	0.1	274	60/36	93%
	Reverse primer 5′–3′	GGAATTCTAGCACCCATCAAAT				
*ftsH* 7	Forward primer 5′–3′	CTGTATCTGGTTCTGAATTTGA	0.5	161	60/36	90%
	Reverse primer 5′–3′	TTGAGATTCTCCTGAAAAACC				
*ftsH* 8	Forward primer 5′–3′	TTGTTGAAAGGTATGTCGGCGTC	0.1	213	56/40	90%
	Reverse primer 5′–3′	TAATAATTCCTTTAGACGAAGAA				
FD-*gyrA*	Forward primer 5′–3′	CTAGAATTGTCGGTGATGTTATGG	0.1	338	64/36	99%
	Reverse primer 5′–3′	AGCCATTCCTACAGCTATACCCG				
FD-*dnaB*	Forward primer 5′–3′	CTTTATCAACCTTTAATAGGTTTAGG	1	355	62/36	99%
	Reverse primer 5′–3′	TTTCTAATATTTTTTGTTCTTCGTCG				

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
