# Peer review of "Flavescence Dorée Phytoplasma Has Multiple ftsH Genes that Are Differentially Expressed in Plants and Insects"

_ijms, 2019, doi:10.3390/ijms21010150_

Round 1

Reviewer 1 Report

In the manuscript „Flavescence dorée phytoplasma has multiple ftsH genes that are differentially expressed in plants and insects” the authors present an overview of the presence of different ftsH gene copies in the FD phytoplasma genome. With their in silico approach they aim to construct a topology model of ftsH integration in the phytoplasma membrane. However, they were not able to generate a satisfying model as they also state themselves. In an experimental approach they analyzed the differential expression pattern of the eight ftsH genes in model and natural hosts of FD phytoplasma. From these results they draw conclusions about a potential role of FtsH in the insect or the plant host, respectively.

The manuscript is very well written and structured. The authors do not provide many new insights from an experimental point of view, but they present interesting models/hypotheses about the potential function of FtsH integrating their results with those from current scientific literature. I appreciate the critical discussion about the potential and limit of prediction software for functional analyses. Information about ftsH sequence differences or differential expression patterns between FD phytoplasma strains is missing. It would be interesting to see if there is a correlation between certain ftsH variants or expression levels and FD virulence or vector transmissibility.

Minor critics

Line

52        before the word “linear” a space is missing

102      correct fstH1 to ftsH1

181      grey not gray

264      headline: … are more expressed in…

239      headline: … is more expressed in…

294      …associated with protein quality…”

430      the fraction bar is off-center

Author Response

Review public FTSH December 2019 Jollard

Reviewer 1

Open Review

English language and style

( ) Extensive editing of English language and style required
( ) Moderate English changes required
(x) English language and style are fine/minor spell check required
( ) I don't feel qualified to judge about the English language and style

Yes

Can be improved

Must be improved

Not applicable

Does the introduction provide sufficient background and include all relevant references?

(x)

( )

( )

( )

Is the research design appropriate?

( )

(x)

( )

( )

Are the methods adequately described?

(x)

( )

( )

( )

Are the results clearly presented?

(x)

( )

( )

( )

Are the conclusions supported by the results?

(x)

( )

( )

( )

Comments and Suggestions for Authors

In the manuscript „Flavescence dorée phytoplasma has multiple ftsH genes that are differentially expressed in plants and insects” the authors present an overview of the presence of different ftsH gene copies in the FD phytoplasma genome. With their in silico approach they aim to construct a topology model of ftsH integration in the phytoplasma membrane. However, they were not able to generate a satisfying model as they also state themselves. In an experimental approach they analyzed the differential expression pattern of the eight ftsH genes in model and natural hosts of FD phytoplasma. From these results they draw conclusions about a potential role of FtsH in the insect or the plant host, respectively.

 The manuscript is very well written and structured. The authors do not provide many new insights from an experimental point of view, but they present interesting models/hypotheses about the potential function of FtsH integrating their results with those from current scientific literature. I appreciate the critical discussion about the potential and limit of prediction software for functional analyses. Information about ftsH sequence differences or differential expression patterns between FD phytoplasma strains is missing. It would be interesting to see if there is a correlation between certain ftsH variants or expression levels and FD virulence or vector transmissibility.

Re: We are very grateful to the reviewer for the helpful comments, especially for the suggestions regarding possible correlations with virulence to plant or transmissibility by insects. We unfortunately have not yet such evidences but are currently very much looking for it.

 Minor critics

 Line

52        before the word “linear” a space is missing ok done

102      correct fstH1 to ftsH1 ok done

181      grey not gray ok done

264      headline: … are more expressed in… ok, “highly” was removed

239      headline: … is more expressed in… ok, “highly” was removed

294      …associated with protein quality…” ok, “increased” was removed

430      the fraction bar is off-center ok, the fraction bar was recentered

Submission Date

02 December 2019

Date of this review

16 Dec 2019 11:42:51

Reviewer 2 Report

The authors present data describing the proteins, phylogeny and gene expression of a  disease of grapes that is transmitted by an insect vector.  The authors use the results to place the bacteria within a phylogentic tree, and suggest possible location of these key proteins in the membrane based on inferences from topology and function. They also explore in some detail the differential response of the bacteria within plant and insect hosts.

The type of work is beyond my expertise, so I did not review in detail the methods. However, the results could be of interest to a larger audience, but as written they are confusing. There are detials that are not noted in the abstract or introduction that should be to bring greater clarity to the logic of the experiments and the importance of these data.

The biggest revision would be to explain why the 'experimental system'  (line 191) included the broad bean plant and Euscelidius variegatus. I'm assuming that is because the plant is not known to suffer from the disease and the insect is not known to vector the disease. But maybe more thought then that was used to select these two.  I would like to see that thought descrbided in the introduction or at least in the results.  Where the authors expecting the results that they observed in figures 4 and 5 given the chose of the plant and insect?  I’d like to know a bit more about the ecology of this ‘experimental system’ to appreciate what can be inferred from these results.

Overall, there are a lot of details, but little synthesis of these details to help the reader understand the need to do these experiments.  I would have liked a hypothesis, especially for the ‘experimental system’.

There are some minor edits that should be addressed, listed below.

ftsH and is used to describe the genes and FtsH to describe the products of those genes.   I noticed in figure 2 its FTSH proteins.  Is that something different? In figure 3, ftsH is used, but so is FtsH. Are they the same thing? Also, is ftsH different from ftsH? If they mean the same thing, can the same name be used throughout?

Line 13- please define the acryonm FtsH when first used in the text.

Line 29- what does gravevine yellow mean? Is that the common name of the disease or class of disease to which it belongs?

Line 191-why was the second insect included in this study? Its is not mentioned in the abstract or the introduction?

What purpose does it serve? Is it not considered a host of the disease? Does it not normally feed on the plants? Please explain.

Lines 194-196: it is very hard to interpret these data. Can they be made into a figure?

Line 223: Unclear. Do you mean within that host? Why not compare across host?

Lines 249-265: I am not sure what the point of this first paragraph is. There are many descriptive sentences but no synthesis of the information given.

Line 320- Are you proposing that ftsH genes are responsible host-vector relationships?

Line 354-357 and Figure 6: If topology is required to determine function and topology could not be determined from these data, then why speculate here?

Author Response

Reviewer 2

Open Review

English language and style

( ) Extensive editing of English language and style required
( ) Moderate English changes required
(x) English language and style are fine/minor spell check required
( ) I don't feel qualified to judge about the English language and style

Yes

Can be improved

Must be improved

Not applicable

Does the introduction provide sufficient background and include all relevant references?

( )

( )

(x)

( )

Is the research design appropriate?

(x)

( )

( )

( )

Are the methods adequately described?

(x)

( )

( )

( )

Are the results clearly presented?

( )

(x)

( )

( )

Are the conclusions supported by the results?

( )

(x)

( )

( )

Comments and Suggestions for Authors

The authors present data describing the proteins, phylogeny and gene expression of a disease of grapes that is transmitted by an insect vector.  The authors use the results to place the bacteria within a phylogentic tree, and suggest possible location of these key proteins in the membrane based on inferences from topology and function. They also explore in some detail the differential response of the bacteria within plant and insect hosts.

The type of work is beyond my expertise, so I did not review in detail the methods. However, the results could be of interest to a larger audience, but as written they are confusing. There are detials that are not noted in the abstract or introduction that should be to bring greater clarity to the logic of the experiments and the importance of these data.

The biggest revision would be to explain why the 'experimental system'  (line 191) included the broad bean plant and Euscelidius variegatus. I'm assuming that is because the plant is not known to suffer from the disease and the insect is not known to vector the disease. But maybe more thought then that was used to select these two.  I would like to see that thought descrbided in the introduction or at least in the results.  Where the authors expecting the results that they observed in figures 4 and 5 given the chose of the plant and insect?  I’d like to know a bit more about the ecology of this ‘experimental system’ to appreciate what can be inferred from these results.

We would like to thank the reviewers for all the remarks and suggestions that helped us to improve the quality of our manuscript. All remarks and request were considered and our responses and manuscript modifications are listed below.

Re: We agree with the reviewer's remark. Therefore, sentences have been added in the manuscript to explain the presence of the experimental pathosystem.

- Lines 63-66: We also hypothesize that the expression of ftsH genes should be host-dependent. two pathosystems were challenged, the natural pathosystem consisting of grapevine and S. titanus, and an experimental pathosystem in which the phytoplasma strain had been maintained since its transmission to broad bean.

- Lines 190-194: Because most functional studies are done on this experimental system, it appeared of interest to determine the gene expression both in the natural and in the experimental pathosystems. Indeed, comparing the two pathosystems allowed us to determine if there was any differences in the expression of ftsH genes in the natural hosts as compared to the experimental ones.

- Lines 372-374: FD is a quarantine disease subjected to mandatory removing of infected grapevine in vineyard. Moreover, the natural insect vector, S.titanus cannot be grown in colony. In order to infect grapevine with FDP, we used an experimental pathosystem as source for FDP.

Our first goal was to determine the gene expression in the natural pathosystem (grapevine-Scaphoideus) to check whether ftsH genes were expressed in both plant and insect vector or if they were more expressed in one specific host (insect or plant). This part of the study aimed at investigating whether ftsH could have a function during a specific step of the phytoplasma cycle (genes involved during insect infection or/and during plant infection). In high confinement greenhouse (FD is a quarantine disease) we infected grapevine via S. titanus to obtain infected plants (Eveillard et al., 2016). We could infect S. titanus (collected on wood in fields) by feeding on infected-broad bean from our collection in greenhouse. Then infected S. titanus could transmit the FDP to the grapevine. However S. titanus cannot be reared in greenhouse, thus limiting the experimental possibilities.

Our experimental pathosystem allows to get FDP-infected plants all year long. Indeed, in contrast to the natural vector S. titanus, the experimental vector E. variegatus can be reared. A collection of FDP in broad bean (the experimental host) is available in our greenhouses. Broad beans are infected via infected-E. variegatus fed on infected-broad bean. Because most functional studies are done on this experimental system, it seemed important to determine the gene expression both in the natural and in the experimental pathosystems. Indeed, comparing the two pathosystems allowed us to determine if there was any specificity of expression of ftsH genes in the natural hosts as compared to the experimental ones. The results (difference of expression or not) were unpredictable

Overall, there are a lot of details, but little synthesis of these details to help the reader understand the need to do these experiments.  I would have liked a hypothesis, especially for the ‘experimental system’.

Re: We understand the reviewer’s remark. A sentence was added to clarify our hypothesis.

- Lines 332-335: One could consider that FDP has to adapt its FtsH machinery (i) to variations of its own proteinic composition associated with host change and (ii) to variations in proteinic composition of the different hosts. One way to achieve such adaptation is to express different FtsH with different specificities.

Comments:

Based on our results, ftsH are expressed in E. variegatus as it is in S. titanus, meaning that the FDP express its ftsH similarly in both insects. On the contrary, ftsH are differentially expressed in grapevine as compared to broad bean. Thus FDP may adapt the ftsH level of expression to its environment in plants. This result confirms the need for comparison of different pathosystems for ftsH expression studies. In future studies, lack of difference of expression in different insects will have to be confirmed by testing additional insect vectors.

There are some minor edits that should be addressed, listed below.

ftsH and is used to describe the genes and FtsH to describe the products of those genes.   I noticed in figure 2 its FTSH proteins.  Is that something different? In figure 3, ftsH is used, but so is FtsH. Are they the same thing? Also, is ftsH different from ftsH? If they mean the same thing, can the same name be used throughout?

Re : We agree with the reviewer ‘s remark. Correction have been made by replacing FTSH by FtsH, and ftsH by ftsH, in text, figures and tables.

Line 13- please define the acryonm FtsH when first used in the text.

Re: We agree with the reviewer ‘s remark. The complete name of FtsH was added Line 47: filament temperature-sensitive protein

Line 29- what does gravevine yellow mean? Is that the common name of the disease or class of disease to which it belongs?

Re: We understand the reviewer’s remark and the sentence has been modified

- Lines 30-32: Symptoms associated to FDP are leaf discoloration (yellowing or reddening) and downward rolling, incomplete lignification of canes and shriveling of grapes and tendrils

Comments:

 Line 191-why was the second insect included in this study? Its is not mentioned in the abstract or the introduction?

What purpose does it serve? Is it not considered a host of the disease? Does it not normally feed on the plants? Please explain.

Re: We agree with the reviewer ‘s remark and sentences were added to clarify:

- Line 63-66 to specify which pathosystems were used.

- Lines 190-194 to justify the choice of these two pathosystems.

- Line 372-374 in the M&M explaining why we need the experimental pathosystem and what is its role.

Comments:

Indeed, “FD is a quarantine disease implying the mandatory removing of infected grapevine in vineyard. Moreover, the natural insect vector, S.titanus is not available in greenhouse. In order to have FDP in grapevine all year long, we used an experimental pathosystem as source of FDP”.

 Lines 194-196: it is very hard to interpret these data. Can they be made into a figure?

Re: We understand the point made by the reviewer. All the results are shown on figure 4 as mentioned line 203. All the genes were expressed (they all show bars of expression). Only one gene was not expressed in insects: see last bar in pink in E. variegatus (EV) and in S. titanus (ST) which is very small, meaning there is no or almost no expression of this gene in insects.

Line 223: Unclear. Do you mean within that host? Why not compare across host?

Re: The reviewer’s remark is right, the term “grapevine” was missing. This was added line 229 and 230.

Comments

This paragraph describes what is observed in grapevine and S. titanus. The next one describes what is observed in broad bean and E. variegatus and at the end of this paragraph (L246-249), there is a comparison between natural and experimental pathosystems.

 Lines 249-265: I am not sure what the point of this first paragraph is. There are many descriptive sentences but no synthesis of the information given.

Re: We agree with the reviewer’s comment. However, little is known about FtsH in phytoplasmas. In the first part of this paragraph, we explain the situation observed for the other known FtsH in other phytoplasmas. It is important to note that for FDP, the situation is different as those already observed as mentioned Line 267-268, i.e. the elevated number of ftsH genes in the FDP genome may not come from Potential Mobile Unit duplication. We don’t have explanation for that yet.

In the second part, for a better clarity to the reader, we expose the reasons why the membrane protein topology is difficult to obtain for phytoplasma which are really special bacteria (low GC content, absence of cleavage of predicted signal peptides….) and it is mentioned Lines 272-276. After these sentences, we propose two conclusions for the in silico analyses which can be important for Phytoplasmologists or researchers working with Mollicutes .

 Line 320- Are you proposing that ftsH genes are responsible host-vector relationships?

Re: We understand the reviewer’s remark and a sentence was added to clarify.

            - Lines 332-335: One could consider that FDP has to adapt its FtsH machinery (i) to variations of its own proteinic composition associated with host change and (ii) to variations in proteinic composition of the different hosts. One way to achieve such adaptation is to express different FtsH with different specificities.

Comments:

We show in this work that the FDP response to its plant environment results in differential expression of the different ftsH, while there is no difference of ftsH expression between the two insect-vectors of the two pathosystems. In this work, the fstH expression was separately studied in insects and in plants and we currently have no clue allowing us to link the observed expression differences to insect-plant interactions.

 Line 354-357 and Figure 6: If topology is required to determine function and topology could not be determined from these data, then why speculate here?

Re: We understand the concern of the reviewer. Indeed, in this part, 2 functional models are proposed, depending on the topology of the protein. Given the small size of the phytoplasma genome, and considering the evolution towards genome reduction, it is expected that these highly duplicated genes may have an essential function either for survival or for pathogenicity. For the scientific community interested in vector-born plant diseases or in adaptation of minimal bacteria (having a reduced genome size) to their environment, we considered it essential to propose in this article functional models for these proteins, taking into account the metabolic specificities of these bacteria.

A sentence was added in this way Lines 255-256: It is then expected that these highly duplicated genes may have an essential function either for survival or for pathogenicity.